# Structural Insights for the Stronger Ability of Shrimp Ferritin to Coordinate with Heavy Metal Ions as Compared to Human H-Chain Ferritin

**DOI:** 10.3390/ijms22157859

**Published:** 2021-07-23

**Authors:** Yingjie Wang, Jiachen Zang, Chengtao Wang, Xiuqing Zhang, Guanghua Zhao

**Affiliations:** 1College of Food Science & Nutritional Engineering, China Agricultural University, Key Laboratory of Functional Dairy, Ministry of Education, Beijing 100083, China; yingjiewang@cau.edu.cn (Y.W.); zangjiachen@cau.edu.cn (J.Z.); 2Beijing Engineering and Technology Research Center of Food Additives, Beijing Technology & Business University, No. 11 Fucheng Road, Haidian District, Beijing 100048, China; wangchengtao@th.btbu.edu.cn

**Keywords:** heavy metal ion removal, prawn ferritin, human H-chain ferritin, cysteine

## Abstract

Although apoferritin has been widely utilized as a new class of natural protein nanovehicles for encapsulation and delivery of nutraceuticals, its ability to remove metal heavy ions has yet to be explored. In this study, for the first time, we demonstrated that the ferritin from kuruma prawns (*Marsupenaeus japonicus*), named MjF, has a pronouncedly larger ability to resist denaturation induced by Cd^2+^ and Hg^2+^ as compared to its analogue, human H-chain ferritin (HuHF), despite the fact that these two proteins share a high similarity in protein structure. Treatment of HuHF with Cd^2+^ or Hg^2+^ at a metal ion/protein shell ratio of 100/1 resulted in marked protein aggregation, while the MjF solution was kept constantly clear upon treatment with Cd^2+^ and Hg^2+^ at different protein shell/metal ion ratios (50/1, 100/1, 250/1, 500/1, 1000/1, and 2500/1). Structural comparison analyses in conjunction with the newly solved crystal structure of the complex of MjF plus Cd^2+^ or Hg^2+^ revealed that cysteine (Cys) is a major residue responsible for such binding, and that the large difference in the ability to resist denaturation induced by these two heavy metal ions between MjF and HuHF is mainly derived from the different positions of Cys residues in these two proteins; namely, Cys residues in HuHF are located on the outer surface, while Cys residues from MjF are buried within the protein shell. All of these findings raise the high possibility that prawn ferritin, as a food-derived protein, could be developed into a novel bio-template to remove heavy metal ions from contaminated food systems.

## 1. Introduction

Heavy metal ion contamination has become one of the most serious environmental problems in recent years and is closely related to food safety because of their non-biodegradable and easily accumulated characters. Heavy metal ions are able to be accumulated in the human body by the food chain and may induce a severe influence on human health [1,2,3]. Since heavy metal ions are the most widespread and harmful contaminants, there are numerous available technologies for water purification and heavy metal ion elimination. These technologies can be simply divided into chemical and biological methods. Chemical methods include adsorption [4], ion-exchange [5], precipitation [6], membrane filtration (microfiltration, nanofiltration, and ultrafiltration), and electrochemical detection [6,7,8]. Unfortunately, these approaches often have many limitations such as requiring complicated sample-preparation processes and harsh reaction conditions, which can introduce byproducts and security issues. The bio-absorption of heavy metals from drinking water and foods is a relatively new technique that has been shown to be promising for the removal of heavy metal ions [5,9,10]. Biological materials with such activities include eggshell-membrane [11], polyacrylic hydrogel [12], and biochars [13]. Despite these materials having effective adsorption and an inexpensive price, they have rarely been developed to detect and remove the heavy metal ions in beverages and milk [14,15] because it is difficult to remove the heavy metal ions without impairing their nutritive contents. Protein as one major biomacromolecule has been explored as a variety of platforms for the synthesis of various nanomaterials to detect and remove heavy metal ions. So far, to the best of our knowledge, there has been no report on the bio-absorption of heavy metals by protein nanocages.

Ferritin represents a class of natural protein nanocages, which is an iron storage protein that widely exists in nature [16]. Ferritin consists of 24 subunits that self-assemble into a 432-point symmetrical cage-like structure with 12 of the two-fold, 8 of the 3-fold, and 6 of the 4-fold channels. The main interactions promoting subunit assembly into a nanocage structure exist in the *C*_2_ interfaces and *C*_3_–*C*_4_ interfaces between the 3-fold and 4-fold axes [17]. The hydrophilic 3-fold channels allow Fe^2+^ and Cu^2+^ ions to enter into the cage, but the narrow, hydrophobic *C*_4_ channels are not permissible to ions [18,19,20]. The ferritin nanocage structure endows it with unique functions: the primary function of ferritin is to store iron while playing an important role in detoxification of iron and oxygen-related radical products in vivo. One ferritin can store up to 4500 iron atoms per protein shell, which corresponds to its holo form. Upon removal of iron from holoferritin under the help of reductants such as ascorbic acid [21], holoferritin reverts to its apo form. Due to its high water-solubility, high stability for thermal and chemical treatments, and unique disassembly or reassembly property controlled by pH, apoferritin has also been utilized as both a nanovehicle for encapsulation and delivery of nutraceuticals and a nanoreactor for the synthesis of nanomaterials [22].

Although ferritin from legume seeds has been extensively studied, less attention has been paid to ferritin from marine products. It has been known that prawn and its products have high a nutritional value and it represents one of the most favored foods for humans. Recently, we established the method for purification of *Marsupenaeus japonicus* ferritin (MjF) and solved its crystal structure at a high resolution of 1.16 Å [23]. Whether MjF has a different physico-chemical property or function from known ferritins remains to be determined. In this study, we demonstrated that MjF exhibits a markedly higher binding capacity to heavy metal ions such as Cd^2+^ and Hg^2+^ as compared to human H-chain ferritin (HuHF), although they share high structural similarity in protein structure. We obtained crystals of two complexes of MjF with these two heavy metal ions, respectively, and solved their crystal structures. These results can explain the large difference between heavy metal binding capacity between MjF and HuHF from a structural viewpoint (Figure 1). These new findings are beneficial for our understanding of the relationship between the structure and function of protein. The present study provides a foundation for developing a biological material for the removal of heavy metal ions, which might be utilized in food and other related industries.

## 2. Results and Discussion

### 2.1. Preparation and Characterization of HuHF and MjF

MjF and HuHF were prepared and purified by ion-exchange and gel-filtration chromatography according to our reported methods [23,24]. Protein fractionation in each purification step was resolved by SDS-PAGE and Native PAGE. After purification, two different ferritin samples were first analyzed by native PAGE, which revealed a single band for both samples, indicating that these two proteins were purified to homogeneity (Figure 2a, lanes 1 and 2). Subsequently, SDS-PAGE was used to analyze the subunit composition of these two samples, and results also showed that there was only one band for both samples (Figure 2b) with an apparent molecular weight of about 19,000 Da for MjF (Lane 2) and approximately 21,000 Da for HuHF (Lane 1), suggesting that HuHF and MjF are both composed of one subunit. To characterize the shape and size, HuHF and MjF molecules were analyzed by transmission electron microscope (TEM). Results showed that HuHF and MjF molecules have a similar spherical shape and are homogeneous in size, and that the exterior diameter (~12 nm) of MjF is the same as that of HuHF (Figure 2c,d). All of these findings are in good agreement with our recent results [23,24], indicating that these two ferritins have been prepared successfully.

### 2.2. Morphological and Aggregation Characterization of MjF and HuHF upon Treatment with Hg^2+^ or Cd^2+^

As is well-known, protein molecules are denatured under some extreme conditions, and thus the structure of the protein is destroyed, finally affecting their function. Like strong acids/alkali, heavy metals are also one of the main factors that cause protein denaturation and aggregation. It has been established that ferritin are robust nanocages, and thus they can withstand temperatures up to 80 °C for 10 min, pH ranges from 3.0–10.0, and exposure to a high concentration of denaturant such as guanidine [25,26]. Inspired by the living environment of prawns, which is usually rich in different metal ions, we wondered whether MjF molecules have a higher stability against denaturation induced by heavy metal ions such as Cd^2+^ and Hg^2+^. If so, MjF nanocages could be developed into a new class of potential biomaterials used for the removal of these two heavy metal ions. To this end, Cd^2+^ and Hg^2+^ were mixed with MjF molecules at different metal/protein ratios of 50/1 100/1, 250/1, 500/1, 1000/1, and 2500/1, respectively, followed by monitoring protein aggregation by transmission electron microscope (TEM). Under the same experimental conditions, HuHF was used as a control sample because it shares high similarity with MjF in amino acid sequence (~62%) analyzed by BLAST, and both of them belong to animal ferritin (Figure 3a). The structure around the 3-fold channel of MjF and HuHF is shown in Figure 3b. The 3-fold channel of HuHF is composed of two acidic side chains, Asp132 and Glu135, both of which are highly conserved among the ferritin family. Differently, the 3-fold symmetry channel of MjF is lined with Glu135, but the highly conserved Asp132 of HuHF is replaced by Lys 132 in MjF. This channel still has a negatively charged surface because of the acidic residues Asp136 substituted with Thr 136 in HuHF [23].

Subsequently, the effect of Cd^2+^ and Hg^2+^ on MjF and HuHF association was studied, respectively. As shown in Figure 4a,c, at a fixed protein concentration of (2.0 μM), MjF molecules constantly stay in a monodispersed by with increasing the concentration of Cd^2+^ or Hg^2+^ from a metal ion/protein ratio of 50/1 to that of 2500/1. In contrast, upon addition of Cd^2+^ to HuHF at a metal ion/protein ratio of 60/1, protein molecules aggregated into larger species under the same experimental conditions (Figure 5a,b). Similarly, upon addition of Hg^2+^ to HuHF at a metal ion/protein ratio of 100/1, protein aggregation also happened (Figure 5d,e); the higher the concentration of the heavy metal ions, the larger the protein aggregation degree (Appendix A). Such a large difference between MjF and HuHF suggests that MjF exhibits a much larger stability against denaturation induced by heavy metal ions as compared to HuHF.

To confirm whether the above observed change in the protein aggregation induced by heavy metal ions also exists in solution, dynamic light scattering (DLS) analyses were carried out, and the results are displayed in Figure 4b,d. It is evident that most of MjF species exhibit a monodispersed distribution with a hydrodynamic radius (*R_H_*) centered at ~8.0 nm upon mixing protein with Cd^2+^ or Hg^2+^, respectively, indicating that protein aggregation hardly occurs in solution. In contrast, large protein aggregated species occurred with HuHF samples under identical conditions (Figure 5c,f). These results are in good agreement with the above results obtained by TEM.

### 2.3. Thermodynamic Characteristics of Cd^2+^ and Hg^2+^ Binding to MjF

The large stability of MjF against denaturation induced by Cd^2+^ and Hg^2+^ suggests that MjF has markedly strong binding activity to these two heavy metal ions. To confirm this idea, isothermal titration calorimetry (ITC), which is an emerging and powerful technique that is widely used to measure the thermodynamic properties of any chemical reaction initiated by the addition of a binding component, was used to study the interaction of MjF with Cd^2+^ and Hg^2+^. Figure 6a shows the raw ITC data for a titration of MjF with Cd^2+^ in MOPS buffer at pH 7.0. The integrated heats for each injection are shown after subtraction of the control injection in buffer alone. The peaks seen in Figure 6a correspond to an endothermic reaction for Cd^2+^ binding to the MjF. The experimental thermodynamic parameters were obtained from the curve-fitting of the integrated heats in Figure 6b to a model of independent binding sites, giving an apparent equilibrium constant *K_app_* = 1.56 × 10^4^ M^−1^ with a Cd^2+^/MjF ratio of about 72 to 1. Since there are 24 subunits per ferritin molecule, these findings suggest that each subunit may capture three Cd^2+^ in solution. The negative value of ΔG (−23.94 kJ mol^−1^) indicates that the binding reaction of Cd^2+^ and MjF occurs spontaneously.

Differently, the peaks seen in Figure 6c correspond to an exothermic reaction for Hg^2+^ binding to the MjF molecule. The experimental thermodynamic parameters through the curve-fitting of the integrated heats in Figure 6d to a model of independent binding sites produced an apparent equilibrium constant *K_app_* = 3.46 × 10^4^ M^−1^ with the reaction Hg^2+^/MjF stoichiometry of about 220 to 1. This result suggests that each subunit can bind with nine Hg^2+^ in solution under the present condition. Thus, it appears that the binding capacity of MjF to Hg^2+^ is around three times larger than that of MjF to Cd^2+^. Moreover, the value of ΔG for the binding reaction of Hg^2+^ to MjF is −25.91 kJ mol^−1^, which is indicative of a spontaneous reaction. In contrast, under the same experimental conditions, titration of Hg^2+^ or Cd^2+^ to HuHF resulted in significant protein aggregation and precipitation, thereby preventing from obtaining a good curve-fitting of the integrated heats. All these findings demonstrate that MjF molecules are able to bind to both Hg^2+^ and Cd^2+^ with high capacity. Possible binding sites for these two heavy metal ions to MjF might consist of the 3-fold channels, the ferroxidase sites of the protein, and other unidentified binding sites that exist of these residues for a metal-binding residue such as His and Asp [27,28]. It has been established that the hydrophilic 3-fold channels are major pathways for metal ions to enter into the inner cavity [16,27], so we compared the size of the channels of MjF with that of HuHF based on their crystal structure and found that the size of the 3-fold channels of MjF is very similar to that of HuHF. Thus, the above difference in the ability to resist denaturation induced by these two heavy metal ions between these two different ferritins is not related to the size of the 3-fold channels.

### 2.4. The Crystal Structure of the Complexes of MjF Plus Cd^2+^ or Hg^2+^

To obtain direct evidence for the binding of Hg^2+^ and Cd^2+^ to MjF, we tried to crystallize MjF upon mixing with heavy metal ions (Hg^2+^ and Cd^2+^, respectively) by screening a wide range of solution conditions at 20 °C. Eventually, we were able to crystallize both complexes of MjF plus Hg^2+^ and MjF plus Cd^2+^ with good diffraction and solved their structure at a resolution of 2.3 Å (I4 space group) and 3.0 Å (P6322 space group), respectively.

The refined structure revealed that a total of 24 Hg^2+^ are bound to MjF nanocage through a metal coordination bond. The binding sites are located on the N terminal of the A-helix at the 3-fold axis channels of MjF, where Cys13 and Asn123 are involved in coordination with Hg^2+^ as shown in Figure 7a. Hg^2+^ binds to the thiolate groups of Cys13 with a distance of 2.4 Å, while it binds to Asn123 with a distance of 2.2 Å. We also found positive electron density map around the ferroxidase site, but it is not clear that the electron density is occupied by Hg^2+^. The side chain of His63, which is an indispensable member of the ferroxidase site, has double conformation—it may be accessed to bind with heavy metal ions. To confirm whether Hg^2+^ binds to the ferroxidase site, we also obtained the crystal structure of MjF where a tiny amount of iron had been removed before the addition of Hg^2+^ according to the reported method [29]. Consequently, there are no positive electron density maps around the ferroxidase site as shown in Appendix A, suggesting that the ferroxidase center is not responsible for the binding of Hg^2+^ to MjF.

Similarly, the crystal structure of the complex of MjF and Cd^2+^ revealed that there are also 24 Cd^2+^ occurring at the protein shell, and that Cd^2+^ also coordinates with Cys13 but with relatively low occupancies (Figure 7b). Thus, it seems that these two heavy metal ions exhibit higher selectivity for Cys residues in protein. However, Cd^2+^ binds to Asp125 instead of Asn123 which is used for coordination with Hg^2+^. These findings are in contrast with a previous result showing that His and Asp located on the interior surface of recombinant horse L-chain apoferritin are responsible for the coordination with Cd^2+^ [30,31]. These two crystal structures revealed that MjF can bind up to 24 heavy metal ions, these findings being different from the above ITC results. Such difference is most likely derived from different experimental conditions.

The above observed protein aggregation induced by Hg^2+^ and Cd^2+^ prevented us from growing the crystal of the complex of HuHF and Hg^2+^ or Cd^2+^ though a co-incubation method, but the crystal structure of HuHF alone can also provide useful information about the binding mode of HuHF with these two heavy metal ions. Each HuHF subunit consists of three cysteine residues as shown in Figure 3b. The cysteine at position 90 appeared to be critical for the formation of ferritin aggregates because it is located at the end of the BC-loop, and thus is completely exposed to the exterior surface of HuHF shell as shown in Figure 3b. This idea is supported by a previous study showing that Cys90 is susceptible to be oxidized and resulting in protein aggregation by disulfide bonds [32]. However, under present conditions, no protein aggregation was formed with HuHF in the absence of Hg^2+^ or Cd^2+^, demonstrating that the observed protein aggregation was derived from protein association induced by these two heavy metal ions. Additionally, Cys102 of HuHF was close to the exterior surface of HuHF, which might play a role in the formation of ferritin aggregates. In contrast, each MjF subunit only contains one Cys residue at position 13, which is located within the 3-fold channel (Figure 3b). Consequently, it is hard for Hg^2+^ or Cd^2+^ to cross-link MjF molecules through its coordination with Cys13 residues due to seriously steric hindrance. Thus, the difference in their ability to resist protein aggregation between MjF and HuHF is most likely derived from the different locations of Cys residues in these two proteins, namely that the Cys residues in HuHF are located on the exterior surface, while Cys residues are buried within the protein shell of MjF. The present results indicate that the coordination mode of heavy metal ions with protein can be controlled by adjusting the position of Cys residues.

The hydrophilic 3-fold channels are believed to be major pathways for iron to enter into the ferritin shell [16,27]. The above crystal structure raises an interesting question of whether the binding of Hg^2+^ to MjF has an effect on iron permeability into the ferritin shell. To answer this question, we compared Fe^2+^ oxidation catalyzed by MjF in the absence and presence of Hg^2+^ when iron ions flux into ferritin with 200 Fe^2+^/protein cage, and results are shown in Figure 7c. It is apparent that the presence of Hg^2+^ dramatically deceased the rate of Fe^2+^ oxidation by oxygen. Thus, it is reasonable to believe that the binding of Hg^2+^ to MjF nearby its 3-fold channels largely impedes the entrance of ferrous ions, thereby inhibiting their fast oxidation at the ferroxidase center. These results demonstrate that the extent of metal ion permeability through the 3-fold channel is variable [33]. As shrimp concentrates heavy metals, it might be supposed that such susceptibility to heavy metal ions would seriously curtail the function of the shrimp ferritin. To shed light on the reason why the binding of Hg^2+^ has such an effect, we compared the crystal structure of MjF in the absence and presence of Hg^2+^. As shown in Figure 7d, the binding of Hg^2+^ to Cys13 and Asn123 did not block the 3-fold channels of MjF directly. However, the presence of Hg^2+^ could neutralize the inherent negative charges occurring at the 3-fold channels, slowing down the rate of iron ions across the channels to reach the ferroxidase center where they are oxidized by oxygen. Thus, it appears that the mechanism by which Hg^2+^ inhibits the oxidation of ferrous ions catalyzed by MjF is different from the mechanism that Hg^2+^ blocks water flows by binding to Cys residues within aquaporin [34,35]. The detail mechanism is under investigation.

## 3. Materials and Methods

### 3.1. Reagents and Chemicals

The gene encoding of HuHF and MjF was built in a pET-3a vector and commercially synthesized by Synbio Technologies (Suzhou, China). BCA Protein Assay Kit, Ampicillin (Amp) and Isopropyl-β-D-thiogalactoside (IPTG) were purchased from sigma (~99%). Tris(hydroxymethyl)methyl aminomethane, 4-Morpholinepropanesulfonic acid (MOPS), sodium chloride, ammonium sulfate, cadmium chloride hydrate, mercury chloride, ferrous sulfate are of analytical grade or above and were generally purchased from Lan Yi and Solarbio (Beijing, China). The solutions were made by directly dissolving the salts in Milli-Q water that was used for all of the experiments.

### 3.2. Apparatus

UV-vis spectra were recorded on a UV spectrophotometer (Varian, 50 Bio, Palo Alto, CA, USA). Transmission Electronic Microscope (TEM) (Hitachi, S-5500, Tokyo, Japan) was used at 80 kV for visualizing the shape and size of ferritin samples. Dynamic light scattering (Wyatt, DynaPro NanoStar, Santa Barbara, CA, USA) was used for hydrodynamic size determination of ferritin samples. All isothermal titration calorimetry (ITC) measurements were carried out using a TA Instruments Nano ITC (TA Instruments, Nano ITC, New Castle, DE, USA).

### 3.3. Preparation of HuHF and MjF

HuHF was purified as previously described [36]. MjF was purified as follows. The *E. coli* strain BL21 (DE3) which contained recombinant vector was grown at 37 °C. 5 mL overnight culture was inoculated into a flask of 500 mL Luria-Bertani/ampicillin medium. Adding 1 mM of isopropyl β-D-1-thiogalactopyranoside to induce protein expression after the cell density reached an absorbance about 0.6 at 600 nm, the cells were harvested by centrifugation at 8000 rpm for 10 min after 8 h of induction. The pellets were resuspended in 20 mM Tris-HCl (pH 8.0) to a concentration of 40 g fresh weight bacteria per liter, followed by sonication to lyse cell (200 watts of power). The supernatant of the resulting crude extract was collected by centrifugation and fractionated by 60% (*w*/*v*) saturation of ammonium sulfate. The deposit was dissolved in 20 mM Tris-HCl (pH 8.0) and dialyzed against the same buffer three times (14.4 kD cut-off). After centrifugation at 10,000 rpm for 20 min and filtration (0.45 μm cut-off), the protein solution was applied to an ion-exchange column (DEAE Fast Flow, GE Healthcare, Chicago, IL, USA), followed by gradient elution with 0–0.5 M NaCl. Finally, the protein solution was concentrated and purified on a gel filtration column (Superdex 300, GE Healthcare), equilibrated with 20 mM Tris-HCl with 150 mM NaCl (pH 8.0). Protein concentrations were determined according to the Lowry method with bovine serum albumin as standard (Line range is 0–0.5 mg/mL with R^2^ = 0.9976, LOQ = 0.005 mg/mL).

### 3.4. SDS-PAGE and Native PAGE

Protein samples were mixed with equal volume of 2× loading buffer, then a 10 µL protein sample (~10 µg of protein) was loaded into each well. It is to note that protein samples for SDS-PAGE experiment need the heat treatment for 5 min in boiling water before loading. The native-PAGE experiment was conducted by using 4–20% polyacrylamide gradient gels running at 120V for 4 h at 4 °C. SDS-PAGE experiment was carried out by using 15% polyacrylamide gels running at 150V for 1 h. Coomassie Brilliant Blue R-250 was used to stain the gels.

### 3.5. Dynamic Light Scattering (DLS) Experiments

Upon treatment of MjF with Hg^2+^ and Cd^2+^, respectively, resulting complexes were analyzed through DLS at 25 °C by a Viscotek model 802 DLS instrument [37]. The Omni SIZE 2.0 software was used to calculate the hydrodynamic radius (*R_H_*) distribution of the prepared samples.

### 3.6. Transmission Electron Microscopy (TEM)

Liquid samples of TEM were diluted with 50 mM MOPS buffer (pH 7.0) and then placed on carbon-coated copper grids. After removing the excess solution with filter paper, 2% uranyl acetate was used to stain ferritin samples, which had been treated with Hg^2+^ and Cd^2+^ for 5 min, respectively. TEM images were obtained through a HitachiH-7650 transmission electron microscope at 80 kV.

### 3.7. Binding Assay of Ferritin to Cd^2+^ and Hg^2+^ by Isothermal Titration Calorimetry (ITC)

The concentration of protein samples was adjusted to 2.0 μM in 50 mM MOPS buffer (pH 7.0). Before each titration, all solutions were degassed thoroughly under vacuum. All isothermal titration calorimetry (ITC) measurements were carried out at 25 °C using a TA Instruments Nano ITC. The solution in the sample cell was stirred at 300 rpm to ensure rapid mixing of the titrant upon injection. As shown in Figure 6 of ITC measurements, the top panel shows the raw ITC data, where each spike represents the injection of titrant into the cell, and the bottom panel shows the integrated and normalized heat plotted. The horizontal coordinate represents the molar ratio of titrant (syringe) to titrand (cell). A background titration, consisting of the same titrate solution but only the buffer solution in the sample cell, was subtracted from each experimental titration to account for the heat of dilution. The reported experimental values were the average of individual best-fit values, fitting with the TA NanoAnalyze software according to the “independent” model which assumes n binding “sites”, each with identical binding properties, but does not account for speciation [38,39].

### 3.8. Crystallization, Data Collection, and Structure Determination

The complex of MjF plus Hg^2+^ or Cd^2+^ was concentrated to 10.0 mg/mL in 20 mM MOPS at pH 7.0. Crystals of the complexes were obtained by using the hanging drop vapor diffusion method. Mixing equal volumes of the mixture of MjF plus Hg^2+^ or MjF plus Cd^2+^ with mother liquid (0.1 M HEPES, pH 7.5; 2.0 M NH_4_HCO_2/_0.1 M C_2_H_9_NaO_5_, pH 4.6; 2.0 M NaCl) produced the complexes crystals at 20 °C. After flash cooled with 25% (*v*/*v*) ethylene glycol as a cryo-protectant, crystals were immediately frozen in liquid N_2_. X-ray diffraction data of the complexes of MjF plus Hg^2+^ and MjF plus Cd^2+^ were collected at resolutions of 2.4 Å and 3.0 Å (Appendix A) at SSRF (BL19U) and then processed, merged, and scaled with the HKL-2000. The structure of the complexes was solved with molecular replacement by MOLREP program in the CCP4 program package using the MjF (PDB code 6A4U) as a search model. Structure refinement was carried out using the Refmac5 program in the CCP4 program package and PHENIX software. The structure was rebuilt using COOT and its figures were produced using PYMOL.

### 3.9. Fe^2+^ Oxidative Deposition in Ferritin

Spectroscopic titration experiments were performed using a Varian Cary 50 spectrophotometer (Varian, USA). The concentration of MjF was 1.0 μM in 20 mM Mops pH 7.0. The titrations were conducted using 1.00 mL of protein and 20 μL of FeSO_4_ (10.0 mM) for mixture at 25 °C [40].

## 4. Conclusions

Heavy metal ion contamination has become one of the most serious environmental problems, which is closely related to food safety. Therefore, it is of great importance to establish a potential method to remove heavy metal ions. In this work, we found that MjF exhibits a much higher ability to resist protein aggregation induced by Cd^2+^ and Hg^2+^, as compared to its analogue, HuHF. The crystal structure of the complexes of MjF and Cd^2+^ or Hg^2+^ revealed that these two heavy metals show higher selectivity for Cys residue. The large difference in the ability to resist denaturation related to Cd^2+^ and Hg^2+^ stems from the different locations of Cys residues in HuHF and MjF. As compared to HuHF, MjF exhibits markedly higher binding activity to these two metal ions with high capacity, raising the possibility that MjF represents a potential food-derived protein for the removal of heavy metal ions from contaminated foodstuffs.

## Figures and Tables

**Figure 1 ijms-22-07859-f001:**
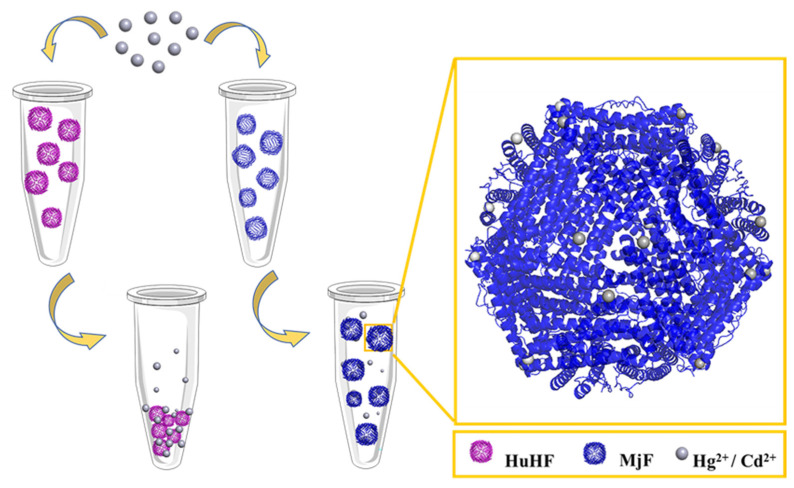
The usual ability of *Marsupenaeus japonicus* ferritin (MjF) nanocages to resist denaturation induced by Cd^2+^ and Hg^2+^ as compared to its analogue, human H chain ferritin (HuHF), under the same experimental conditions.

**Figure 2 ijms-22-07859-f002:**
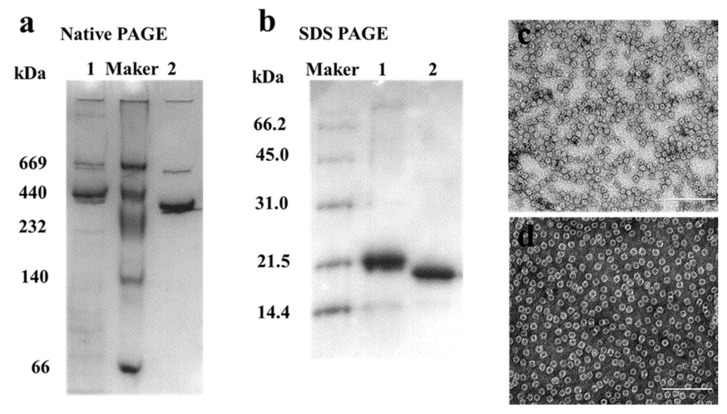
The characterization of HuHF and MjF. (**a**) Native PAGE analyses for HuHF (Lane 1) and MjF (Lane 2). (**b**) SDS-PAGE analyses for HuHF (Lane 1) and MjF (Lane 2). (**c**,**d**) are the TEM images of MjF and HuHF proteins, respectively.

**Figure 3 ijms-22-07859-f003:**
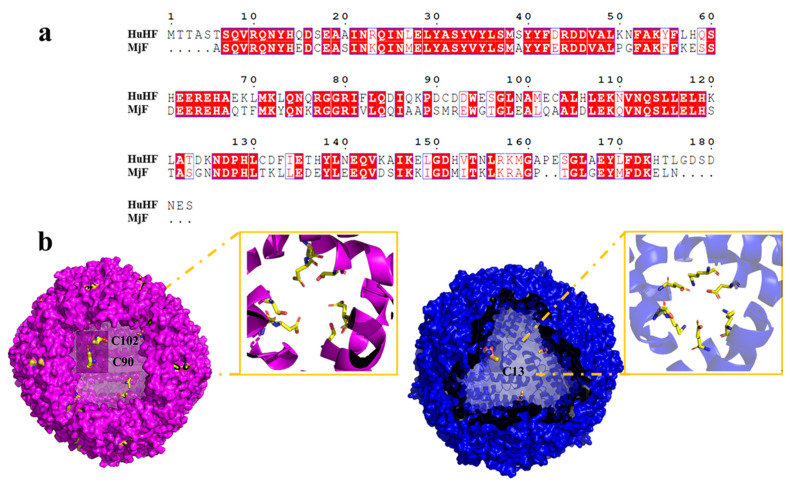
Comparison of HuHF and MjF structure. (**a**) Complete sequences alignment of HuHF and MjF revealed that MjF and HuHF share high similarity in the primary structure. Homologous residues are highlighted in red boxes. (**b**) Both HuHF and MjF have a similar shell-like structure. Cys residues are highlighted in both proteins. Additionally, the structure of the 3-fold channels of MjF and HuHF was compared.

**Figure 4 ijms-22-07859-f004:**
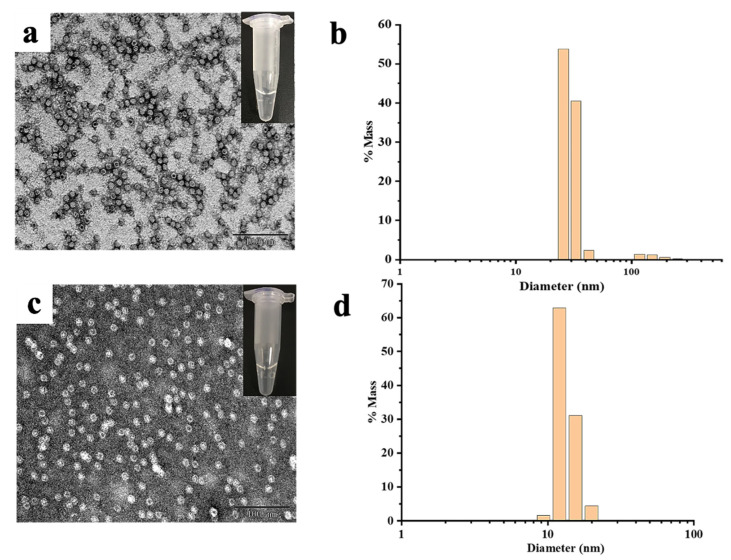
(**a**) TEM images of the MjF-Cd^2+^. (**b**) DLS images of the MjF-Cd^2+^. (**c**) TEM images of the MjF-Hg^2+^. (**d**) DLS images of the MjF-Hg^2+^. Inserts show the solution simples of MjF-M^2+^.

**Figure 5 ijms-22-07859-f005:**
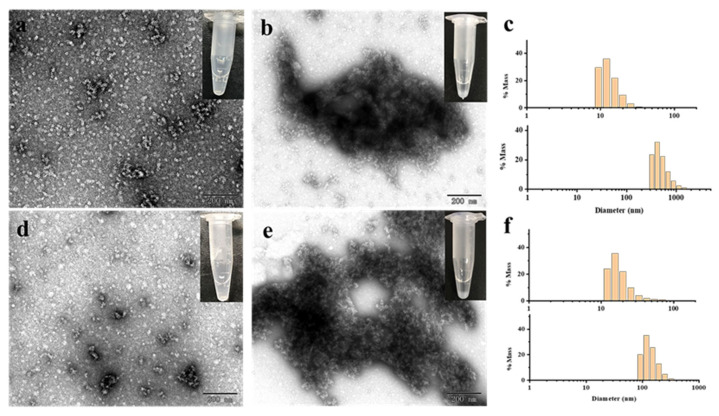
(**a**,**b**) TEM images of HuHF upon treated with Cd^2+^ at a metal/protein ratio of 50:1 and 60:1, respectively. (**c**) DLS analyses of HuHF upon treated with Cd^2+^ at a metal/protein ratio of 50:1 (the panel on top) and 60:1 (the panel at the bottom), respectively. (**d**,**e**) TEM images of HuHF treated with Hg^2+^ at a metal/protein ratio of 60:1 and 100:1, respectively. (**f**) DLS analyses of HuHF upon treated with Hg^2+^ at a metal/protein ratio of 60:1 (the panel on top) and 100:1 (the panel at the bottom), respectively.

**Figure 6 ijms-22-07859-f006:**
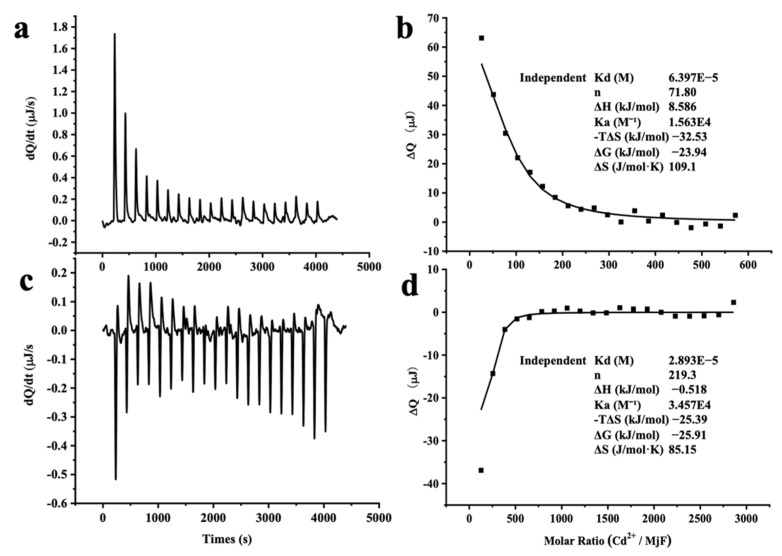
ITC measurement of the binding of M^2+^ to MjF in 50 mM MOPS buffer (pH 7.0). (**a**,**c**) Raw data obtained for continuous injection of 2.5 μL solution to 200 μL MjF protein solution. (**a**) 4 mM Cd^2+^ to 2.0 μM MjF protein solution. (**c**) 20 mM Hg^2+^ to 2.0 μM MjF protein solution. (**b**,**d**) Titration plot derived from the integrated heats of binding of (**a**,**c**), corrected for the heat of dilution.

**Figure 7 ijms-22-07859-f007:**
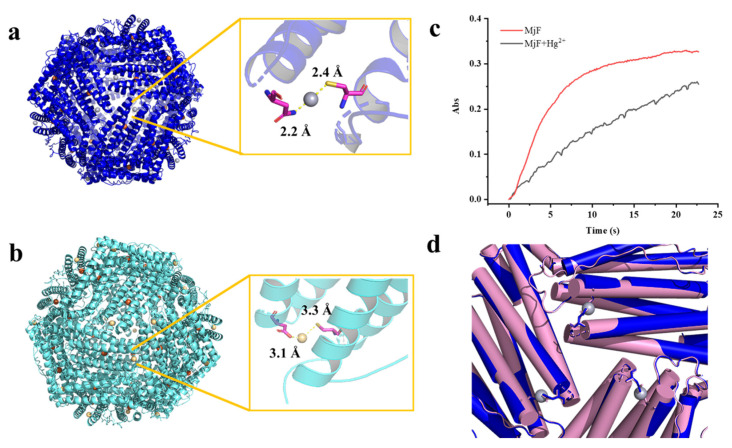
The crystal structure of the complex of MjF plus Hg^2+^ or Cd^2+^. (**a**) Residues Cys13 and Asn123 in MjF are responsible for its binding to Hg^2+^. (**b**) Residues Cys13 and Asp125 in MjF are responsible for its binding to Cd^2+^. (**c**) Comparison of kinetic curves for Fe^2+^ oxidation, catalyzed by MjF in the absence and presence of Hg^2+^. Conditions: 1.0 μM MjF in 20 mM Mops (pH 7.0), 200 μM FeSO_4_, 25 °C, 1000 μM HgCl_2_. (**d**) Structure alignment of MjF in the presence (blue) and absence (pink) of Hg^2+^ indicate that the 3-fold channels are very similar.

## Data Availability

Data available upon request from corresponding author.

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
