# Peer review of "Structural Insights for the Stronger Ability of Shrimp Ferritin to Coordinate with Heavy Metal Ions as Compared to Human H-Chain Ferritin"

_ijms, 2021, doi:10.3390/ijms22157859_

Round 1
Reviewer 1 Report
The manuscript needs a some revisions.
I recommend that you be given experimental details of the methods used in the analyzes.
I recommend highlighting the importance of the protein study in the removal of cadmium and mercury as contaminante.

Author Response
Comments and Suggestions: I recommend that you be given experimental details of the methods used in the analyzes.
Response: Suggestions were followed in the revised version.
Comments and Suggestions: I recommend highlighting the importance of the protein study in the removal of cadmium and mercury as contaminante.
Response: Suggestions were followed in the revised version.
(1) Why HRMS were not used for these purposes. Macromolecules characterization should be carried out by high resolution mass spectrometry? Can you give an adequate explanation?
Response: Thanks for your suggestions. We used Transmission Electronic Microscope (TEM) to characterize the shape and size of ferritin because it can give us the typical characteristics of 24 mer ferritin. In previous studies, we found that the high resolution mass spectrometry was not a good methods because native ferritins have 24 subunits (~480 kDa) that out of the test range of HRMS.
(2) Oxidation state of each metal and its ionic strength propertie can cause the precipitation, also pH and temperature, etc.
Response: Thanks for your suggestions. The ionic strength properties, pH, temperature of heavy metal ions buffers was consistent with the protein buffers. Therefore, we can exclude the oxidation state of each metal causing the precipitation.
(3) C102, C90 and C13, can you report other place or improve the visibility?
Response: Thanks for your suggestions. Previous researches mentioned that Cys is the main amino acid binding with heavy metal ions. Hence, we hypothesized that C102, C90 of HuHF and C13 of MjF may bind with heavy metal ions and marked out these places.
Reviewer 2 Report
This is an interesting paper. It might be useful to comment on the size of the threefold and fourfold spaces between the MjF and HuF since the threefold channels are almost certainly important routes for iron and copper inflows and outflows to and from the central core. References should be included.
1) If the authors want MjF to be used as a commercial source for metal ion removal it would be important to establish if heavy metals can be transported into the ferritin core as well as bind to Cys residues....
2) References should be included line 60 re iron mobilization by reductants.
3) It would be useful to confirm whether the authors hypothesis that Co causes human ferritin aggregation by disulfide linkages using strong reducing agents such as dithiosulfide.
I have made some suggested changes to the text in red see attached.

Author Response
It might be useful to comment on the size of the threefold and fourfold spaces between the MjF and HuF since the threefold channels are almost certainly important routes for iron and copper inflows and outflows to and from the central core. References should be included
Response: Thanks for your suggestions. Suggestions were followed in the revised version. The size of the threefold and fourfold spaces of MjF is similar to HuF through the comparison of HuF and MjF structure.
(1) If the authors want MjF to be used as a commercial source for metal ion removal it would be important to establish if heavy metals can be transported into the ferritin core as well as bind to Cys residues.
Response: Thanks for helpful suggestions. It is important to establish other methods to transport heavy metals into the ferritin core because we have not found the binding sits in ferritin core in the crystal structures of MjF.
(2) References should be included line 60 re iron mobilization by reductants.
Response: Suggestions were followed in the revised version.
(3) It would be useful to confirm whether the authors hypothesis that Co causes human ferritin aggregation by disulfide linkages using strong reducing agents such as dithiosulfide.
Response: Thanks for your suggestions. In this study, we introduced heavy metal ions has coordination interactions with Cys of ferritin, reducing agents could not reduce aggregation, therefore human ferritin aggregation was not caused by disulfide linkages.
Round 2
Reviewer 1 Report
The observations have been corrected and the manuscript is ready to be published.Author Response
The observations have been corrected and the manuscript is ready to be published.
Thank you for your helpful suggestions.
Reviewer 2 Report
The edits have improved the paper slightly, however although the authors mention that there is a Cys residue in the threefold channels of MjF as illustrated in Fig 7, it would be useful to have the sequence comparisons of MjF and HuF of the threefold channels described more fully.
Although the superficial appearance of these channels may look similar a more detailed examination would be useful. I would expect that Hg would bind to and block the MjF threefold channels by binding irreversibly with Cys15. much in the same way as it blocks water flows by binding to cys residues within aquaporin. see Agre, et al Aquaporin CHIP: the archetypal molecular water channel. Am. J. Physiol. 265 (Renal Fluid Electrolyte Physiol. 34): F463-F476, 1993.
Thus as the shrimp appears to concentrate heavy metals, it might be expected that any such susceptibility to Hg would seriously curtail its function, in contrast to HuF. Some comments on this might be worthwhile. Also as I previously hinted it might be useful to determine the extent that the channel metal ion permeability is variable as has recently been described see L.E. Johnson, T. Wilkinson, P. Arosio, A. Melman, F. Bou-Abdallah, Effect of chaotropes on the kinetics of iron release from ferritin by flavin nucleotides, Biochim. Biophys. Acta - Gen. Subj. 1861 (2017) 3257–3262.
Better still demonstrate experimentally the effects or lack of effects or organic mercurials on ferritin iron permeability.
Author Response
(1) It would be useful to have the sequence comparisons of MjF and HuF of the threefold channels described more fully.
Response: All suggestions were followed in the revised manuscript. We have added this information to the revised version.
(2) Although the superficial appearance of these channels may look similar a more detailed examination would be useful. I would expect that Hg would bind to and block the MjF threefold channels by binding irreversibly with Cys15. Much in the same way as it blocks water flows by binding to cys residues within aquaporin. much in the same way as it blocks water flows by binding to cys residues within aquaporin. See Agre, et al Aquaporin CHIP: the archetypal molecular water channel. Am. J. Physiol. 265 (Renal Fluid Electrolyte Physiol. 34): F463-F476, 1993.
Response: Thanks for helpful suggestions. To elucidate whether the binding of Hg2+ to MjF has an effect on iron permeability into ferritin shell, we compared Fe2+ oxidation catalyzed by MjF in the absence and presence of Hg2+ when iron ions flux into ferritin with 200 Fe2+/ protein cage. We found that the presence of Hg2+ in MjF dramatically deceased the rate of Fe2+ oxidation by oxygen. Thus, it is reasonable to believe that the binding of Hg2+ to MjF nearby its 3-fold channels largely impedes the entrance of ferrous ions, thereby inhibiting their fast oxidation at the ferroxidase center. To shed light on the reason why the binding of Hg2+ has such effect, we compared the crystal structure of MjF in the absence and presence of Hg2+. As shown in Figure 7d, the binding of Hg2+ to Cys13 and Asn123 did not block the 3-fold channels of MjF directly. However, the presence of Hg2+ could neutralizes the inherent negative charges occurring at the 3-fold channels, slowing down the rate of iron ions to across the channels to reach the ferroxidase center where they are oxidized by oxygen. Thus, it appears that the mechanism by which Hg2+ inhibits the oxidation of ferrous ions catalyzed by MjF is different from the mechanism that Hg2+ blocks water flows by binding to Cys residues within aquaporin. We added these new results as Figure 7c, d to the revised manuscript, and we also these discussion to the revised version.
(3) Thus, as the shrimp appears to concentrate heavy metals, it might be expected that any such susceptibility to Hg would seriously curtail its function, in contrast to HuF. Some comments on this might be worthwhile. Also as I previously hinted it might be useful to determine the extent that the channel metal ion permeability is variable as has recently been described see L.E. Johnson, T. Wilkinson, P. Arosio, A. Melman, F. Bou-Abdallah, Effect of chaotropes on the kinetics of iron release from ferritin by flavin nucleotides, Biochim. Biophys. Acta - Gen. Subj. 1861 (2017) 3257–3262.
Response: Suggestions were followed in the revised version. Our new results (Fig. 7c,d) showed that the presence of Hg2+ in MjF dramatically deceased the rate of Fe2+ oxidation by oxygen. These results demonstrate that the extent of metal ion permeability through the 3-fold channel is variable.
Additionally, as the shrimp concentrates heavy metals, such susceptibility to heavy metal ions would seriously curtail the function of the shrimp ferritin. These information and corresponding references have been added to the revised manuscript.
(4) Better still demonstrate experimentally the effects or lack of effects or organic mercurials on ferritin iron permeability.
Response: Thanks for helpful suggestions. Our new results as Figure 7c.d showed that the binding of heavy metal ions to MjF has a negative effect on Fe2+ oxidation catalyzed by shrimp ferritin.
Round 3
Reviewer 2 Report
no further comments.